# Roundup in the Reproduction of Crucian Carp (*Carassius carassius*): An In Vitro Effect on the Pituitary Gland and Ovary

**DOI:** 10.3390/ani13010105

**Published:** 2022-12-27

**Authors:** Magdalena Socha, Joanna Szczygieł, Jarosław Chyb, Ewa Drąg-Kozak, Mirosława Sokołowska-Mikołajczyk, Elżbieta Brzuska, Anna Pecio, Małgorzata Grzesiak

**Affiliations:** 1Department of Animal Physiology and Endocrinology, University of Agriculture in Krakow, Al. Mickiewicza 24/28, 30-059 Krakow, Poland; 2Institute of Ichthyobiology and Aquaculture in Gołysz, Polish Academy of Sciences, Zaborze, Kalinowa 2, 43-520 Chybie, Poland; 3Department of Animal Nutrition and Biotechnology, and Fisheries, University of Agriculture in Krakow, Al. Mickiewicza 24/28, 30-059 Krakow, Poland; 4Department of Comparative Anatomy, Institute of Zoology and Biomedical Research, Jagiellonian University in Krakow, Gronostajowa 9, 30-387 Krakow, Poland; 5Department of Endocrinology, Institute of Zoology and Biomedical Research, Jagiellonian University in Krakow, Gronostajowa 9, 30-387 Krakow, Poland

**Keywords:** Roundup, oocytes, ovulation, pituitary, reproduction, crucian carp, GVBD, fish

## Abstract

**Simple Summary:**

Roundup is a widely used herbicide. Its active component, glyphosate, is described as an endocrine disrupter. The effect of this glyphosate-containing herbicide on the reproductive parameters of crucian carp (*Carassius carassius*) was investigated in this study. The entire pituitary glands and ovarian fragments of crucian carp were incubated in vitro in the medium containing two different concentrations of Roundup (1 and 10 ng/mL). We assessed the final oocyte maturation and ovulation, structural changes in follicles, levels of 17,20β-progesterone, and luteinizing hormone. The mRNA levels of zona radiata (chorion) genes, kisspeptin, and its receptor as well as luteinizing hormone and estrogen receptors were also determined. Obtained results indicate that Roundup may adversely affect oocyte maturation and the quality of crucian carp eggs, suggesting that exposure to this herbicide can lead to reproductive disorders in fish.

**Abstract:**

Roundup, the most popular herbicide in global agriculture, is regarded as an endocrine disruptor causing alterations of important hormones at the hypothalamic−pituitary−gonadal axis as well as impairment of gametogenesis. The whole pituitary glands of crucian carp (*Carassius carassius*) were incubated for 3 h in the medium containing Roundup (0-control, 1 and 10 ng/mL). The level of luteinizing hormone (LH), and mRNA transcript abundance of kisspeptin (*kiss-1*) and its receptor (*gpr54*), were determined. The isolated ovarian fragments were incubated for 24 h in the presence of Roundup and the following effects on reproductive parameters were determined: the final oocyte maturation and ovulation, structural changes in follicles, secretion of 17,20β-progesterone (17,20β-P) as well as mRNA transcript abundance of the luteinizing hormone receptor (*lhr*), estrogen receptors (*erα*, *erβ1*, *erβ2*), and zona radiata (chorion) proteins (*zp2* and *zp3*). Roundup inhibited final oocyte maturation and decreased the percentage of ovulated eggs, and furthermore, caused structural changes in the ovarian follicular components. There were no significant changes in the measured hormone levels and analyzed genes mRNA transcript abundance. Summing up, obtained results indicate that Roundup may adversely affect oocyte maturation and the quality of eggs, suggesting that exposure to this herbicide can lead to reproductive disorders in fish.

## 1. Introduction

Aquatic exposure to glyphosate based-herbicides (GBHs), among them Roundup, has shown many adverse reproductive disorders in both male and female fish [1]. There are data showing a reduction in egg production [2], decreased sperm motility [3,4], and changes in egg swelling [4,5]. Roundup induced developmental and physiological effects in fish after early life exposure to this herbicide [4,6,7,8,9,10].

Warningly, these widely used chemicals (GBHs/Roundup) of environmental concern are detected not only in a freshwater environment [9,11,12] but also in tissues of non-target aquatic and terrestrial organisms [13,14], and the levels of their/GBHs residuals are escalating [15].

Up to now, the following actions of Roundup or its active ingredient glyphosate as endocrine disruptors, in mammals have been demonstrated: changes in the activity of steroidogenic enzymes [16,17,18], alteration in levels of important reproductive hormones such as FSH, LH, testosterone, and estrogen [19,20,21], as well as impairment of gamete maturation [18,21,22,23,24]. These results indicate the possibility of the direct action of Roundup/glyphosate at all levels of the hypothalamo−pituitary−gonadal axis (HPG axis).

As in other vertebrates, fish reproduction is also regulated by the HPG axis. The key neuroendocrine hormone kisspeptin (Kiss), binding to its receptor GPR54- initiates the production of the gonadotropin-releasing hormone (GnRH) within the hypothalamus. GnRH activates the synthesis and release of the gonadotropin follicle-stimulating hormone (FSH) and luteinizing hormone (LH) from the pituitary [25,26,27]. Gonadotropins and their receptors (FSH-R, LH-R) play an essential role in the process of gametogenesis and in the regulation of the production of sex steroid hormones such as testosterone (T), 11-ketotestosterone (11-KT), estradiol (E2), and progestogens-17,20β-dihydroxy-4-pregnen-3-one (17,20β-P) [26,28]. In female fish, E2 stimulates oogonial proliferation, vitellogenesis, choriogenesis, and 17,20β-P, known as the maturating inducing steroid (MIS), promotes initiation of germ cell meiosis, and follicular maturation and ovulation [29,30]. During final oocyte maturation (FOM), endocytosis (incorporation of yolk proteins as vitellogenins, choriogenins, fat-soluble vitamins, and lipids into the oocyte) is terminated, the resumption of meiosis starts and the germinal vesicle migrates toward the micropyle, and next the disintegration of the germinal vesicle membrane, known as GVBD (germinal vesicle breakdown), occurs. At this stage, oocytes absorb water and inflate, the pressure within the follicle increases, the follicular wall is ruptured, and the oocyte ovulates and becomes an egg [31]. The proper levels of LH, 17,20β-P, and activation of the maturation-promoting factor (MPF), a complex consisting of existing cdc2-kinase and newly synthesized cyclin B are necessary for induction of resumption of meiosis and final follicular maturation [28,30]. Very often, this critical stage of oocyte maturation, relevant to reproductive success, is affected by environmental toxicants acting as hormone mimics, receptor blockers, and/or enzyme inhibitors or other stressful factors, which affect fish breeding in captivity/captive-reared fish [26,28,31,32,33].

Oocytes of vertebrates are surrounded by coats composed of several glycoproteins, named differentially (the chorion in fish or zona pellucida in mammals) despite similar structure and function [34]. The zona pellucida is important for embryo protection during development. Protein factors on its surface might also play a significant role in sperm−egg interactions. From two layers, which make a fish chorion, a thick inner and a thin outer, the first one is suggested to be homologous to zona pellucida [35]. In teleost, the synthesis of glycoproteins composing this layer was considered to take place in the ovary or/and in the liver. Recent studies indicate that proteins of zona pellucida can also be synthesized in other tissues [36]. The zona pellucida zp2 and zp3 genes were present during all developmental stages of ovaries in carp [37] and crucian carp [38]. Their level decreases during the late stages of oocyte maturation. In Nile tilapia, zp genes that were expressed in the ovary and liver were expressed equally with oocyte growth [39]. In this study, the mRNA levels of two zp genes, zp2 and zp3, were analyzed by RT-qPCR using RNA extracted from ovaries exposed to Roundup.

There is evidence showing the adverse effect of glyphosate or GBHs on zebrafish ovarian maturation and final oocyte maturation [33,40,41]. Maskey et al. [33] have demonstrated that zebrafish (*Danio rerio*) ovarian follicles incubated with glyphosate exhibited significantly lower maturation rates measured by GVBD. In vivo adult zebrafish females’ exposure to Roundup for 15 days resulted in such ultrastructural changes in the ovary as: vacuolization in follicular cells, increased perivitelline space, and impaired mitochondria. Moreover, important changes in vitelline protein content and a decrease in the diameter and number of late ovarian follicles were observed after Roundup treatment [41]. Furthermore, the studies on Japanese medaka (*Oryzias latipes*) showed that early-life exposure to glyphosate and Roundup might negatively affect the central regulators of reproduction in adulthood, such as the Kiss/Gpr54 system [9].

To our knowledge, there is no data on the reproductive dysfunctions at the level of pituitary and gonads of crucian carp (*Carassius carassius*) during the spawning season in the context of Roundup’s adverse influence. It is worth noting that crucian carp is the native species in Europe with reference to decreasing populations in various areas [42,43,44]. Therefore, we hypothesize that Roundup may affect final oocyte maturation and ovulation by the disruption of LH and 17,20β-P levels and transcripts of important reproductive genes in crucian carp. The following main goals were to be determined:the secretion of the luteinizing hormone (LH), mRNA transcript abundance of kisspeptin (kiss-1) and its receptor (gpr54), and estrogen receptors (erα, erβ1, erβ2) after 3-h incubation of the crucian carp whole pituitary glands in medium containing Roundupthe final oocyte maturation and ovulation, structural changes in follicles, secretion of 17,20β-progesterone (17,20β-P) as well as mRNA transcript abundance of luteinizing hormone receptor (lhr), estrogen receptors (erα, erβ1, erβ2) and zona radiata (chorion) proteins (zp2 and zp3) after 24 h during in vitro incubation of crucian carp ovarian fragments in medium containing Roundup.

## 2. Materials and Methods

### 2.1. Pituitary Preparation and Incubation

The experiment was conducted at the end of May on sexually mature crucian carp (*Carassius carassius*) females obtained from the Fishery Station belonging to the Department of Animal Nutrition and Biotechnology, and Fisheries, the University of Agriculture in Kraków, Poland. The use of animals was in accordance with the Act of 15 January 2015 on the Protection of Animals Used for Scientific or Educational Purposes and Directive 2010/63/EU of the European Parliament and the Council of 22 September 2010 on the protection of animals used for scientific purposes. The study was approved by the II Local Institutional Animal Care and Use Committee (IACUC) in Kraków, Poland (resolution No. 263/2015). Pituitaries were obtained from 18 females (the mean body weight 259 ± 0.41 g) after decapitation. Gonad maturity was specified as a percentage of body weight (gonadosomatic index GSI) and was about 16%. Collected pituitary glands were washed twice with preincubation medium containing 2% Ultroser SF (Sepracor S.A., France) and 1% antibiotic-antimycotic (Sigma-Aldrich: Saint Louis, MI, USA) and transferred into a 24-well microplate (Nunc: Roskilde, Denmark). Each well contained 2 pituitaries in 2 mL of pre-incubation medium. Then the plates were covered/sealed and incubated for 1 h at 22 °C for equilibration. After the pre-incubation period medium was replaced (after washing) with medium containing Roundup (Roundup 360 SL, with glyphosate at a concentration of 360 g/L—Monsanto Europe S.A./N.V) at concentrations: 0 (control), 1, and 10 ng/mL medium (R1, R10, respectively) and incubated for 3 h at 22 °C. Each treatment group consisted of three wells with two pituitaries in each. At the end of the incubation period, the media were collected and frozen at −20 °C until LH determination by ELISA [45]. The pituitaries were sampled and fixed with RNAlater reagent until RNA isolation was performed.

### 2.2. Follicle Preparation and Oocytes Maturation

Ovaries were collected from five sexually mature crucian carp females (selected from the first experiment) and placed in chilled Cortland’s medium. Follicles were separated by gently pipetting the fragments of ovaries, washed with medium, and transferred into 3 24-well microplates, coated with Cortland’s medium with 1% BSA (Sigma-Aldrich). Each well contained 0.5 mL of follicles (~150 ovarian follicles) and 1.5 mL medium, including the tested concentration of Roundup: 0 (control), 1, and 10 ng/mL (R1, R10, respectively). The ovarian follicles had completed vitellogenesis and none of them had undergone germinal vesicle breakdown (GVBD). There were 3 experimental groups, each group consisted of 15 wells (the follicles of each female were in triplicate-*n* = 5/treatment per group/triplicates). Then the plates were covered and incubated for 24 h at 22 °C with periodical shaking. At the end of the exposure period, the media were collected and frozen at −20 °C until 17,20β-P determination by ELISA kit for fish 17α,20β dihydroxy progesterone (MyBioSource, Cat. No. MBS2602842). The follicles were fixed in Serr’s fluid (acetic acid–ethanol–formalin mixture 1:6:3 *v*/*v*), then gradually dehydrated with a mixture of ethanol and turpentine oil, and finally, they were immersed in turpentine oil for transparency to visualize cell nuclei. Then the stage of oocyte maturity was determined by examining the position of the germinal vesicle (GV): in the center (I), shifted but not crossing the half of the oocyte radius (II), located peripherally, near micropyle (III), invisible, after GVBD (IV) [46,47]. For histological analysis, the ovarian follicles, after 24 h incubation with Roundup, were fixed in 10% buffered formalin. The eggs from the third repetition were sampled and fixed with RNAlater reagent until RNA isolation was performed.

### 2.3. Total RNA Isolation and cDNA Synthesis

The pituitaries and ovarian follicles were stored in RNAlater at −20 °C. The SV Total RNA Isolation System (Promega) was used for total RNA isolation. RNA was stored at −80 °C until further analysis. cDNA was synthesized using a First Strand cDNA Synthesis Kit (Thermo Scientific). All RT-qPCRs were carried out with an Applied Biosystems 7500 Real-Time PCR System using 5 μL of 10-times diluted cDNA, 9.6 μL of Maxima SYBR Green qPCR Master Mix (Thermo Scientific), 0.2 μL of each primer (Table 1), and 5 μL of nuclease-free water. The RT-qPCR conditions were as described previously [10]. qPCR data analysis was performed using the 2^−ΔΔCt^ method [48] with *β-actin* and *gapdh* genes as an endogenous control using Applied Biosystems 7500 and Microsoft Excel 2016 software.

### 2.4. Hormone Analysis

Incubation media obtained and stored, as described above, were used to assess the quantitative measurements of LH and 17,20β-P levels by ELISA. The LH level was assayed by the method established by Kah et al. [45] for common carp LH with a sensitivity in the range of 0.6–100 ng/mL. The detection range and the Lower Limit of Detection (LLD) of the ELISA kit for fish 17,20β-P were 0.312–20 ng/mL and 0.06 ng/mL, respectively. The procedure was used following the manufacturer’s instructions (MyBioSource, Cat. No. MBS2602842). LH and 17,20β-P concentrations were determined by measuring the absorbance using a 96-well plate reader (BIO-TECH INSTRUMENTS, EL 311) at 490 and 450 nm, respectively.

### 2.5. Ovarian Morphology

Just after incubation with different concentrations of Roundup, ovarian follicles were fixed in 10% buffered formalin for 24 h. Then, samples were dehydrated in an increasing gradient of ethanol (50–100%), cleared in xylene, and embedded in Paraplast (Sigma-Aldrich, MO, USA). Sections of 5 µm in thickness were mounted on slides coated with 3-amino-isopropyl-triethoxysilane (Sigma-Aldrich), deparaffinized and rehydrated through decreasing ethanol solutions (100–50%), and stained with hematoxylin-eosin (HE) by routine protocol. Selected sections were photographed using a Nikon Eclipse NieU microscope and a Nikon Digital DS-Fi1-U3 camera (Nikon, Tokyo, Japan) with corresponding software.

### 2.6. Statistical Analysis

Statistical analysis of LH and 17,20β-P levels were carried out using a nonparametric two-tailed Mann-Whitney *U*-test and GraphPad Prism statistical software (version 5, 2007, GraphPad Software: San Diego, CA, USA). Gene expression data were compared using the Kruskal-Wallis test followed by Conover’s posthoc using MedCalc software (MedCalc v18.11). To determine normality, a D’Agostino-Pearson test was performed. For all tests, differences among means were considered to be statistically significant for *p* ≤ 0.05.

## 3. Results

### 3.1. The Effect of Roundup on the LH and 17,20β-P Levels

There were no significant differences in the level of LH measured in the medium after 3 h incubation with Roundup (1 or 10 ng/mL) as compared to the control group (Figure 1A). After 24 h incubation with Roundup (1 or 10 ng/mL) there were no significant differences in the level 17,20β-P in the incubation medium compared to the control group (Figure 1B).

### 3.2. The Effect of Roundup on the mRNA Transcript Abundance of kiss-1, gpr54, lhr, erα, erβ1, erβ2, zp2, and zp3

There were no significant differences in mRNA transcript abundance of *kiss-1* and *gpr54*, in the crucian carp pituitary gland exposed for three hours to two tested concentrations of Roundup (Figure 2A). Compared to the control, mRNA levels of the above genes were not significantly different in the experimental groups.

No significant changes were also observed in mRNA transcript abundance of *lhr*, *erα*, *erβ1*, and *erβ2* in crucian carp ovarian follicles exposed for 24 h to 2 tested concentrations of Roundup compared to the control group (Figure 2B). Compared to the control, *zp2* and *zp3* mRNA transcript abundance remained unchanged in ovarian follicles of crucian carp from groups exposed to Roundup (Figure 2C).

### 3.3. The Effect of Roundup on Oocyte Maturation and Ovarian Morphology

The observed stages of final oocyte maturity after 24 h of incubation in the presence of Roundup (1 or 10 ng/mL) were changed. After incubation with both tested concentrations of Roundup, the percentage of oocytes with peripherally situated germinal vesicle (III) and the percentage of oocytes after GVBD (IV) were significantly decreased compared to the control (Figure 3). In the case of percentages of oocytes with centrally situated germinal vesicle (I) and shifted (II) but not crossing half of the oocyte radius (II), there were no significant differences between the experimental groups and the control (Figure 3). The results of 24 h treatment with Roundup caused evident structural changes in the ovarian follicular component (Figure 4). After exposure to Roundup, the presence of accumulated cytoplasm under the chorion was more abundant (Figure 4B–D).

## 4. Discussion

To study whether Roundup has any direct role in crucian carp oocyte development, we incubated for 24 h post-vitellogenic follicles in vitro. Our results showed an inhibitory effect of this herbicide on the final maturation of crucian carp oocytes. Both tested concentrations of Roundup lowered the percentage of oocytes with peripherally situated germinal vesicles as well as after GVBD, which confirms the possibility of a direct action of this pesticide at the gonadal level. These results are similar to those obtained by Maskey et al. [33], who reported a significantly lower maturation rate measured by GVBD in zebrafish oocytes incubated with glyphosate, the active ingredient of Roundup. Furthermore, the study by Zhang et al. [23] on mouse oocytes also revealed that glyphosate had reduced rates of germinal vesicle breakdown and first polar body extrusion, probably by generating oxidative stress and early apoptosis. The study by Slaby et al. [52] presented the negative effect of Roundup and glyphosate on amphibian oocyte maturation. The cytological analyses pointed out an increase in the occurrence of abnormalities during spindle morphogenesis of *Xenopus laevis* oocyte maturation after exposure, which confirms the possible adverse influence of GBHs on this essential step in the reproduction of another aquatic species. Our results showed that the incubation with Roundup changed the structure of the ovarian follicular component. The amount of accumulated cytoplasm under the chorion was more evident, as well as the separation between the chorion and the cytoplasm after 24 h of Roundup treatment. Another study on zebrafish with a longer (15 days) in vivo intoxication with Roundup (0.065 and 6.5 mg/L) also resulted in structural changes in the ovary presented as a vacuolization in the follicular cells, an increase of the perivitelline space, and a decrease of vitelline envelope/chorion thickness in the ovarian follicles that may affect the process of endocytosis/exocytosis important in the proper oocyte/eggs development [41]. The changes in the physical properties of the egg surface caused by various adverse environmental factors, such as Roundup, may disturb the water uptake and ion exchange between the perivitelline fluid and the aquatic environment, which in consequence, may also affect egg swelling [5]. In fish, proper egg swelling is very important for successful fertilization as well as further embryonic development. Studies by Ługowska [4,5] have shown the adverse effect of Roundup water contamination on common carp (*Cyprinus carpio* L.) and grass carp (*Ctenopharyngodon idella*) egg swelling. In the case of common carp eggs, the reduction in swelling was noted at 20 and 120 min after fertilization in the groups treated with Roundup (0.1 or 10 mg/L). Furthermore, the observations have shown a decrease in common carp embryo survival [4], which was also reported by Fiorino et al. [7] and Socha et al. [10], pointing to the adverse effect of GBHs on the early life development of some, sensitive cyprinids.

Studies investigating embryonic toxicity depending on the presence or absence of the chorion showed that chorion removal increased sensitivity to chemicals in zebrafish [53,54,55]. Studies by Tran et al. [55] also revealed that in zebrafish embryos exposed to organophosphate esters, the presence of the chorion affects gene expression. However, the toxic effect on the chorionic proteins’ synthesis and chorion permeability and structure can be dose-dependent [56]. The number of widely used chemicals shown as causing endocrine disruption and deregulation of the hypothalamic−pituitary−gonadal axis increased [57]. Little is known about the sensitivity of zp gene expression to anthropogenic chemicals getting into the aquatic environment. In fish, synthetic estrogens like 17α-ethinylestradiol can significantly affect up-regulation of genes related to zona pellucida in gonads of adult topmouth culters (*Culter alburnus*) [36] and Japanese medaka (*Oryzias latipes*) [58]. In our study, despite morphological changes in chorion structure, the gene expression (*zp2* and *zp3*) was unchanged in Roundup-treated oocytes compared to the control.

It has been reported that glyphosate possessed estrogenic activity, inter alia, by the ability to alter estrogen receptor (erα and β) expression in human breast cancer cells [59]. That is why in our study, the relative expression of erα, erβ1, and erβ2 mRNA transcripts were analyzed in crucian carp oocytes after Roundup treatment, but the results showed no important changes in mRNA transcript abundance of estrogen receptors; however, the light upregulation in erα and erβ2 were noticed in both tested concentrations.

In our study, there were no changes in the level of 17,20β-P after 24 h incubation with Roundup. The secretion of MIS to the incubation medium was similar in Roundup-exposed oocytes to the control group. It was in contrast to the results obtained on the Prussian carp (*Carassius gibelio*) post-vitellogenic oocytes incubated with the same Roundup concentrations (1 or 10 ng/mL), in which the following simultaneous changes were observed: the significant decrease in 17,20β-P level and the inhibition in oocytes maturation/ovulation after 24 h [60]. In our study also, the expression of mRNA levels of lhr in Roundup-treated oocytes was not significantly different in comparison to the control, which is reflected by the unchanged 17,20β-P level. It seems that the inhibition of crucian carp final oocyte maturation after Roundup exposure occurred probably through the changes in other factor/factors [61] involved in the FOM process, which needs additional study to clarify this mechanism.

As already mentioned, the critical role in promoting final oocyte maturation is played by the surge of LH and increased follicular sensitivity, which reflects increased lhr mRNA. Previous studies on mammals have shown that exposure to Roundup/glyphosate has changed the level of gonadotropins [19,20]. In our study, the direct action of Roundup on the pituitary gland was observed after short, 3-h in vitro exposure. The secretion of LH to the incubation medium was at a similar, unchanged level after this time between the Roundup-treated groups and the control one. The abundance of mRNA transcripts for genes encoding the *kiss*/*gpr54* system reflects changes occurring in the hypothalamo−pituitary axis controlling the LH secretion by activating the GnRH neurons. In our study, mRNA transcript abundance of *kiss-1*, its receptor-*gpr54*, after 3-h incubation of the crucian carp whole pituitary glands with Roundup was not significantly altered. However, the study by Smith et al. [9] documented that embryonic Roundup exposure caused changes in neurons involved in reproduction at the molecular level in the brain of adult Japanese medaka females. The significant decrease in kisspeptin receptors’ (*gpr54-1* and *gpr54-2*) mRNA levels was reported after Roundup exposure. In males, the adverse effects of Roundup and glyphosate, also at low, environmentally relevant concentrations, were observed at the level of gonads, as a significant decrease in the follicle-stimulating hormone receptor (*fshr*) and androgen receptor alpha (*arα*) mRNA transcript abundances in adult Japanese medaka testes. These results show the possible adverse effects of GBHs on the reproductive system in fish even long after the intoxication in the aquatic environment [9].

## 5. Conclusions

Taken together, obtained results indicate that Roundup may adversely affect final crucian carp oocyte maturation, which can lead to reproductive disorders. Exposure to Roundup decreased the percentage of oocytes with peripherally situated germinal vesicles and the rate of oocytes after GVBD and caused structural changes in the ovarian follicular components. Thus, the Roundup residue in the aquatic environment represents a serious threat to the proper function of the fish reproductive system and further studies are still needed.

## Figures and Tables

**Figure 1 animals-13-00105-f001:**
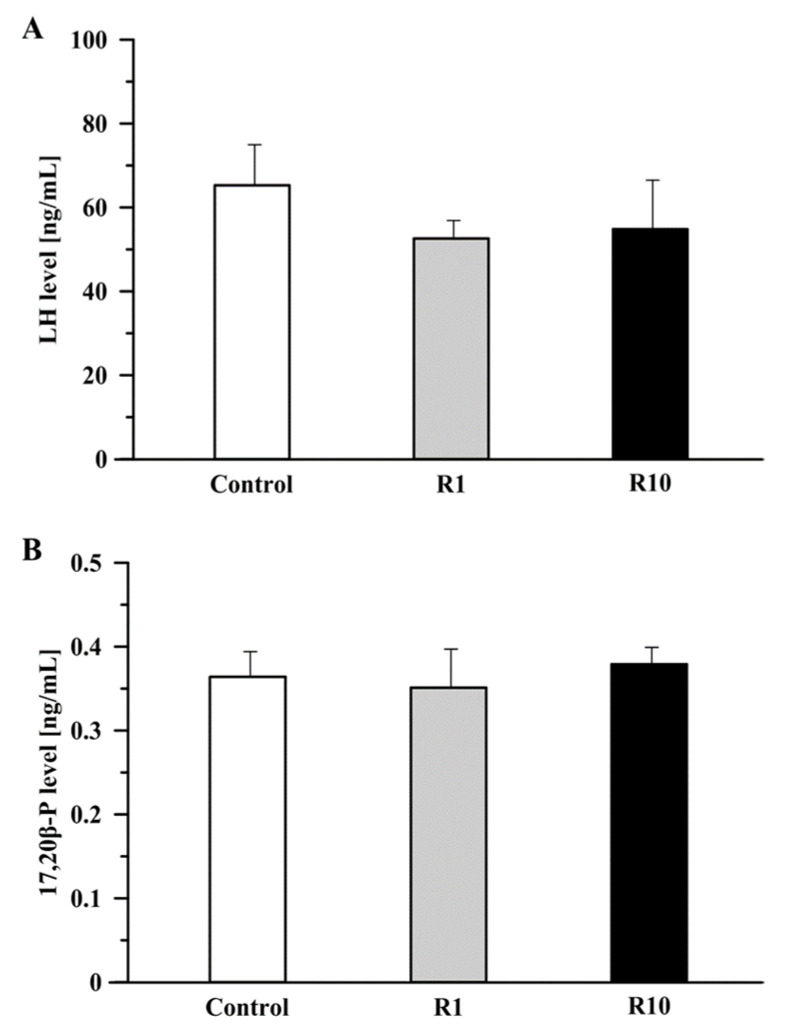
The effect of Roundup at a concentration of 1 (R1) and 10 ng/mL (R10) on LH (**A**) and 17,20β-progesterone (17,20β-P) (**B**) secretion from crucian carp (*Carassius carassius*) pituitary (**A**) and ovarian follicles (**B**) in vitro. Data are expressed as means ± SEM (*n* = 3 wells/each group, 2 pituitaries/well, *n* = 5 wells/each group, ~150 ovarian follicles/well).

**Figure 2 animals-13-00105-f002:**
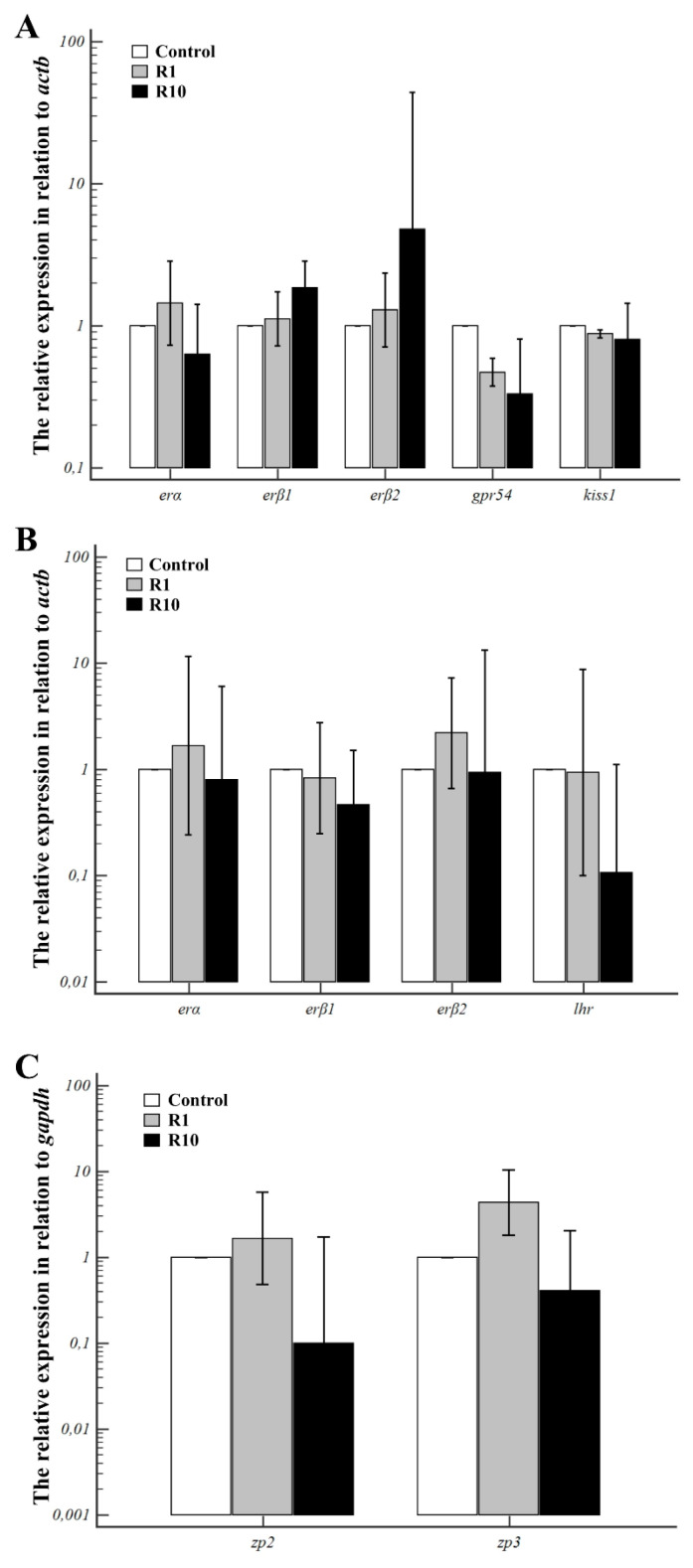
Alternation in *kiss-1*, *gpr54*, *erα*, *erβ1*, *erβ2* mRNA transcript abundance in crucian carp (*Carassius carassius*) pituitary (**A**), *lhr*, *erα*, *erβ1*, *erβ2* and *zp2*, *zp3* mRNA transcript abundance in ovarian follicles (**B**,**C**) exposed to Roundup at a concentration of 1 (R1) and 10 ng/mL (R10). Data were shown as a logarithm of relative expression compared to *β-actin* (**A**) and *gadph* (**B**,**C**) and expressed as means ± SD (*n* = 3 wells/each group, 2 pituitaries/well, *n* = 5 wells/each group, ~150 ovarian follicles/well).

**Figure 3 animals-13-00105-f003:**
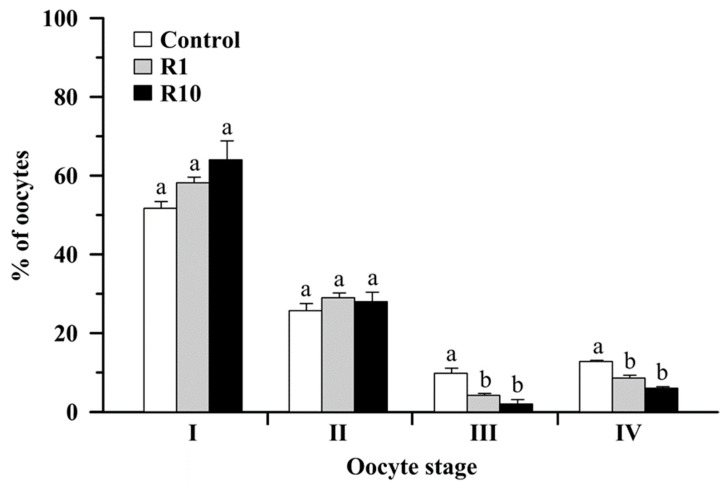
The effect of Roundup at a concentration of 1 (R1) and 10 ng/mL (R10) on the percentage of crucian carp (*Carassius carassius*) oocyte maturity stages after 24 h incubation. Data are expressed as means ± SEM (*n* = 5 wells/per group, ~150 ovarian follicles/well). Different letter superscripts indicate significant differences between the means of experimental groups, *p* ≤ 0.05.

**Figure 4 animals-13-00105-f004:**
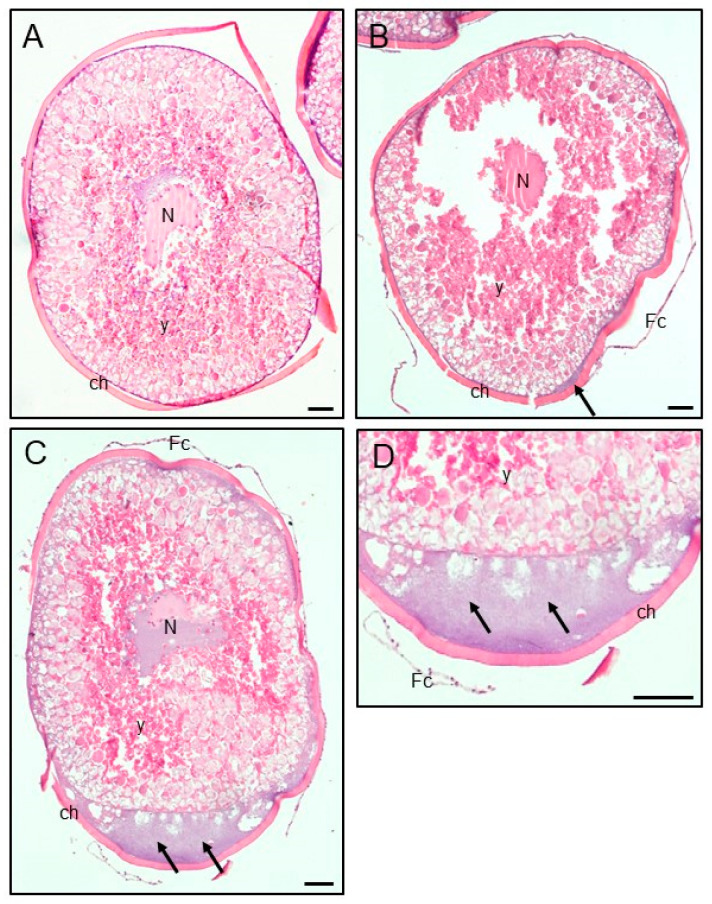
Morphological analysis of crucian carp (*Carassius carassius*) ovarian follicles from the control group (**A**) and those exposed to Roundup at a concentration of 1 (**B**) and 10 ng/mL (**C**,**D**). Arrows indicate the cytoplasm accumulated under the chorion that was more abundant following Roundup treatment (**B**–**D**) than in the control oocytes (**A**). Bar = 50 µm (**A**–**D**) ch, chorion; Fc, follicular cells; N, nucleus; y, yolk.

**Table 1 animals-13-00105-t001:** Sequences of primers used for real-time quantitative PCR analysis.

Gene	Forward Primer (5′-3′)	Reverse Primer (5′-3′)	Reference
*kiss1*	CAGATCCTCAGCGAAACACA	GCAAGCATGTTCTGCTCTCT	[49]
*gpr54*	TTTGGGGACTTCATGTGTCG	ATCTGTGGTGTTCGATGACG	[49]
*lhr*	CCTCTGCATCGGTGTGTATC	TAGACAGATAGTTCGCCGCC	[49]
* erα *	GAGGAAGAGTAGCAGCACTG	GGCTGTGTTTCTGTCGTGAG	[50]
* erβ1 *	GGCAGGATGAGAACAAGTGG	GTAAATCTCGGGTGGCTCTG	[51]
* erβ2 *	GGATTATTCACCACCGCACG	TTCGGACACAGGAGGATGAG	[51]
*β-actin*	GTTTTGCTGGAGATGATGCC	TTCTGTCCCATGCCAACCAT	[49]
*gapdh*	TGATGCTGGTGCCCTGTATGTAGT	TGTCCTGGTTGACTCCCATCACAA	[49]
* zp2 *	CTGAGTCTGGATTCGGTTCA	ATATCCTCCGTCCTCCATCA	
* zp3 *	CCAGCCATCTGGTTTGTCTA	CACAGGGGTGTAGGTAAGAG	

## Data Availability

Not applicable.

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
