# Peer review of "Roundup in the Reproduction of Crucian Carp (Carassius carassius): An In Vitro Effect on the Pituitary Gland and Ovary"

_animals, 2022, doi:10.3390/ani13010105_

Round 1

Reviewer 1 Report

The manuscript is very interesting and is written quite carefully, but the authors did not avoid some ambiguities and inaccuracies, so I am asking for clarification.

1.    Please explain why the following doses of Roundup 1ng and 10ng/mL were used and why the incubation lasted 3h for the pituitary and 24h for the ovary (LH and 17,20 β-P) and 24h for the oocytes.  

2.    I suggest the subsection title in MM: “Ovarian morphology”, not “Morphology”; it will be more informative

3.    Why stage IV is called "after GVBD" (line 175) in MM (Follicle preparation and oocyte maturation) and "ovulated eggs" (line 261) in the results? According to the authors, are these the same stages? According to me, no.

4.    Line 262 should be: Figure 3 (not Figure 2).

5.    Why did the authors call structural changes in a component of the ovarian follicle: ultrastructure? In my opinion, this is not correct, ultrastructural analyzes have not been performed.

6.    Figure 4C was not quoted in the description of the results concerning changes in the structure of the ovarian follicle; it is not really known what figure 4C represents (which dose of Roundup)?

7.    And the most important question: how many oocytes (after incubation with Roundup) with a changed envelope structure were observed? Single oocytes, most of them or all oocytes? This is very important information to assess the real impact of Roundup on oocyte morphology.

8.    Please enter “zona radiata” with the term “chorion”, otherwise lines 129-130 may be incomprehensible.

9.    In my opinion, the Discussion is too long, wordy; there are unnecessary repetitions, e.g. lines 299-303 and lines 325-328.

Author Response

Responses to the Reviewer 1 comments

 We would like to thank You for all helpful comments and suggestions, which we believe will make the article better. The answers to all comments are appended below.

  1. Please explain why the following doses of Roundup 1ng and 10ng/mL were used and why the incubation lasted 3h for the pituitary and 24h for the ovary (LH and 17,20 β-P) and 24h for the oocytes.

In the present paper, the used doses of Roundup (1 and 10 ng/mL, respectively: 0.36 and 3.6 ng/mL of glyphosate acid equivalents) were based on the detected amounts of glyphosate acid equivalents in natural water bodies ranging from 0.01 to 5.2 mg/L (Annet et al., 2014 -https://doi.org/10.1002/jat.2997; Smith et al., 2019 - https://doi.org/10.1016/j.aquatox.2019.03.005). The same, environmentally relevant Roundup concentrations were also tested in our previous work (Socha et al., 2021 - https://doi.org/10.1016/j.theriogenology.2021.09.007).

  1. I suggest the subsection title in MM: “Ovarian morphology”, not “Morphology”; it will be more informative

Thank you for pointing this out, it has been changed (line 278 in revised MS).

  1. Why stage IV is called "after GVBD" (line 175) in MM (Follicle preparation and oocyte maturation) and "ovulated eggs" (line 261) in the results? According to the authors, are these the same stages? According to me, no.

Of course, germinal vesicle breakdown (GVBD) is the very important process occurring just before ovulation. According to Lubzens et al. Oogenesis in teleosts: how eggs are formed. Gen Comp Endocrinol. 2010 Feb 1;165(3):367-89: “At the follicular level, oocyte maturation and ovulation are thus closely integrated and partly overlapping events”.). That is why we often use these words interchangeably, but of course these are not the same stages. It has been changed following your suggestion (line 361 in revised MS)

  1. Line 262 should be: Figure 3 (not Figure 2).

Thank you, it has been changed (line 362 in revised MS).

  1. Why did the authors call structural changes in a component of the ovarian follicle: ultrastructure? In my opinion, this is not correct, ultrastructural analyzes have not been performed.

Thank you for pointing this out, it has been changed, according to the right suggestion (line 161, 365, 447,557 in revised MS).

  1. Figure 4C was not quoted in the description of the results concerning changes in the structure of the ovarian follicle; it is not really known what figure 4C represents (which dose of Roundurepresenrp)?

According to Reviewer 2, Figure 4 has been changed and entire oocytes have been shown. The description of this figure was rewritten with all references in the text. Please see the new Figure 4.

  1. And the most important question: how many oocytes (after incubation with Roundup) with a changed envelope structure were observed? Single oocytes, most of them or all oocytes? This is very important information to assess the real impact of Roundup on oocyte morphology.

Thank you for this important question. For the preparation of histological samples of ovarian follicles, duplicates from each group were used (n=5 wells/each group, ~ 150 ovarian follicles/well). Finally, for histological analysis of each group about 50–60 follicles (10-15 follicles were seen on one slice) were count. We observed respectively 20% and 40% follicles with more abundant accumulated cytoplasm under the chorion in the R1 and R10 groups (Figure 4 B, C and D). It has been added to the manuscript (line 386 in revised MS ).

  1. Please enter “zona radiata” with the term “chorion”, otherwise lines 129-130 may be incomprehensible.

Thank you for this suggestion, it has been changed (line 163 in revised MS).

  1. In my opinion, the Discussion is too long, wordy; there are unnecessary repetitions, e.g. lines 299-303 and lines 325-328.

Thank you for pointing this out. We have shortened the discussion, hopefully, it is better now.

Reviewer 2 Report

The introduction is clearly written and logical, and the hypothesis of this work is clearly stated.  Bibliographic references are relevant. The question is whether the herdicide "Roundup" has an impact on the hypothalamic-pituitary-gonad  (HPG) axis that controls fish reproduction. The experimentis carried out on the crucian carp and concerns the reproduction of the female (Carassius carassius).
Isolated pituitary glands or ovarian follicles taken from mature adult females, are exposed in vitro to "Roundup" .

1° ) It is not clear what justifies Roundup doses used in in vitro exposure.

Why these doses especially? Are these "environmental" doses? Do you have any literature references on environmental doses that would justify the doses used  in vitro that are presented here?

 2°) It would be interesting to be able to compare the results using glyphosate alone to the results using the total formulation of the whole roundup in the present paper.

Material and methods

1- page4, line187 : precise if 0.2µM or 0.2µL of each primer

2- Briefly describe the GVBD process in crucian carp (or carp  species), precising the timing and duration between stage I and stage IV

Results

page 9 : figure 4 :

1- legend : line 279 : "In the Round-up treated groups (C-E)" : the sentence without a verb is incomprehensible.

2-    Pictures : follicles shown in picture A, C, D and E are obviously not at same stage. One can easily see that picture A shows an oocyte with nucleus still localized at its center. One cannot see the nucleus in the pictures C,D and E. The authors assume that the difference is due to the incubation with Round-up. However, we need to see the whole follicle and the oocyte nucleus position to make one's opinion concerning the effect of Round-up.

3- it would be beneficial to show the follicles in their entirety, where the position of the oocyte nucleus is shown, then to zoom in to enlarge certain areas, and to show the plasma accumulation proposed by the authors.

4- Figure 4, -E looks like ovulation: it is necessary to show the nucleus of the oocyte to clarify that it is not an ovulation figure.  (choose another vue of your slice for example)

Discussion

1- Could you dose the "Round-up" concentration in the follicles following the 24h incubation- and also  in the pituitaries following the incubation ? This would be informative of the putative way of action of the contaminant.

2-  Page 10, line 337 to 341 : « the relative expression of era, erb1, erb2 mRNA were analysed in crucian carp oocytes » : correction : the analysis is made on the whole follicle, not only the oocyte .

The discussion takes care to repeat each of the points tested in these incubation experiments. It contains the bibliography in detail. However, it seems to be far too extensive and very detailed in relation to the results obtained. The reader is somewhat disappointed that the results do not include all the parameters set out in the conclusions (biomarkers of ovulation:cyclin, cdc2, calcium; biomarkers of oxidative stress:cat sox, gpx, ....). Discussion goes in all directions, without providing detailed precise information but the reader does not really find the evidence in the results presented.

The conclusion should open the way for further analyses or experiments.

Author Response

Responses to the Reviewer 2 comments

We would like to thank you for all helpful comments and suggestions, which we believe will make the article better. The answers to all comments are appended below.

Comments and Suggestions for Authors

The introduction is clearly written and logical, and the hypothesis of this work is clearly stated.  Bibliographic references are relevant. The question is whether the herdicide "Roundup" has an impact on the hypothalamic-pituitary-gonad  (HPG) axis that controls fish reproduction. The experimentis carried out on the crucian carp and concerns the reproduction of the female (Carassius carassius).
Isolated pituitary glands or ovarian follicles taken from mature adult females, are exposed in vitro to "Roundup" .

1° ) It is not clear what justifies Roundup doses used in in vitro exposure.

Why these doses especially? Are these "environmental" doses? Do you have any literature references on environmental doses that would justify the doses used  in vitro that are presented here?

In the present paper, the used doses of Roundup (1 and 10 ng/mL, respectively: 0.36 and 3.6 ng/mL of glyphosate acid equivalents) were based on the detected amounts of glyphosate acid equivalents in natural water bodies ranging from 0.01 to 5.2 mg/L (Annet et al., 2014 -https://doi.org/10.1002/jat.2997; Smith et al., 2019 - https://doi.org/10.1016/j.aquatox.2019.03.005). The same, environmentally relevant Roundup concentrations were also tested in our previous work (Socha et al., 2021 - https://doi.org/10.1016/j.theriogenology.2021.09.007).

 2°) It would be interesting to be able to compare the results using glyphosate alone to the results using the total formulation of the whole roundup in the present paper.

Yes, we do agree that it would be very interesting to compare the results using pure glyphosate and the whole Roundup formulation. However, in this study we were able to use only commercial formulation of Roundup 360SL.

Material and methods

  • page4, line187 : precise if 0.2µM or 0.2µL of each primer

Thank you for pointing this. It has been written more precisely: 0.2 µL (please see subchapter “Total RNA isolation and cDNA synthesis” in revised MS).

2- Briefly describe the GVBD process in crucian carp (or carp  species), precising the timing and duration between stage I and stage IV

Germinal vesicle breakdown is a dramatic phenomenon within mature oocyte arrested in prophase I of meiotic division. It is the first morphological sign of resumption of meiosis within an oocyte leading to the dissolution of nuclear envelope. The above process lasts between 8 and 10 hours (Yang et. al 1999 - https://doi.org/10.1038/sj.cr.7290012). The duration of the final oocyte maturation, when the migration of germinal vesicle toward the animal pole starts (II, III), and next the disintegration of nucleus membrane (GVBD), takes about 24 h. Of course, when the appropriate levels of all necessary factors (i.e. LH, MIS and MPF) are present. It was shortly described in the Introduction (line 105–109 of revised MS).

Results

page 9 : figure 4 :

1- legend : line 279 : "In the Round-up treated groups (C-E)" : the sentence without a verb is incomprehensible.

Thank you for pointing this, it has been deleted.

2-    Pictures : follicles shown in picture A, C, D and E are obviously not at same stage. One can easily see that picture A shows an oocyte with nucleus still localized at its center. One cannot see the nucleus in the pictures C,D and E. The authors assume that the difference is due to the incubation with Round-up. However, we need to see the whole follicle and the oocyte nucleus position to make one's opinion concerning the effect of Round-up.

Thank you for this suggestion. It has been changed. We have shown the whole follicles in the same stage for all analysed groups (new Figure 4) with the visible oocyte nucleus.

3- it would be beneficial to show the follicles in their entirety, where the position of the oocyte nucleus is shown, then to zoom in to enlarge certain areas, and to show the plasma accumulation proposed by the authors.

It has been changed following your suggestion.

4- Figure 4, -E looks like ovulation: it is necessary to show the nucleus of the oocyte to clarify that it is not an ovulation figure.  (choose another vue of your slice for example)

We have inserted in the Figure 4 C another view (just the follicles in their entirety), where the visible germinal vesicle is situated in the centre of the oocyte. That is why we classify this kind of follicles as not undergone GVBD/ovulation.

Discussion

1- Could you dose the "Round-up" concentration in the follicles following the 24h incubation- and also  in the pituitaries following the incubation ? This would be informative of the putative way of action of the contaminant.

We agree, it would be very interesting. The aim of our study was to observed the “action” of environmentally relevant concentration of Roundup, not its accumulation/concentration in tested tissues. But, taking into consideration the present information about the Glyphosate Based Herbicides and their detection not only in freshwater environment [9,11,12] but also in tissues of non-target aquatic and terrestrial organisms [13,14] it would be interesting to evaluate its content after in vitro exposure in tested tissue.

2-  Page 10, line 337 to 341 : « the relative expression of era, erb1, erb2 mRNA were analysed in crucian carp oocytes » : correction : the analysis is made on the whole follicle, not only the oocyte .

It has been changed (lines  349 in revised MS ).

The discussion takes care to repeat each of the points tested in these incubation experiments. It contains the bibliography in detail. However, it seems to be far too extensive and very detailed in relation to the results obtained. The reader is somewhat disappointed that the results do not include all the parameters set out in the conclusions (biomarkers of ovulation:cyclin, cdc2, calcium; biomarkers of oxidative stress:cat sox, gpx, ....). Discussion goes in all directions, without providing detailed precise information but the reader does not really find the evidence in the results presented. The conclusion should open the way for further analyses or experiments.

Thank you for pointing this out. We have shortened this part of manuscript. We have deleted the information about some not described biomarcers (from line 441 in revised MS), and 2 sentences starting from line 471: “Our results showed that the incubation … cytoplasm after 24 h of Roundup treatment”. Then we deleted the whole paragraph starting from line 485: “It was also reported that glyphosate caused the increase in expression of steroidogenic factor-1 in zebrafish oocytes…. lowered number of viable swim-up fry was detected [60].”). Next we deleted the fragment (from the line 530): „The study conducted by Owagboriaye et al. …, what indicates the plausible glyphosate-mediated interference with their production at the pituitary level [19].”
